# OPRM1 Gene Polymorphism in Women with Alcohol Use Disorder

**DOI:** 10.3390/ijms25053067

**Published:** 2024-03-06

**Authors:** Agnieszka Boroń, Aleksandra Suchanecka, Krzysztof Chmielowiec, Małgorzata Śmiarowska, Jolanta Chmielowiec, Aleksandra Strońska-Pluta, Remigiusz Recław, Anna Grzywacz

**Affiliations:** 1Department of Clinical and Molecular Biochemistry, Pomeranian Medical University in Szczecin, Powstańców Wielkopolskich 72 St., 70-111 Szczecin, Poland; agnieszka.boron@pum.edu.pl; 2Independent Laboratory of Health Promotion, Pomeranian Medical University in Szczecin, Powstańców Wielkopolskich 72 St., 70-111 Szczecin, Poland; o.suchanecka@gmail.com (A.S.); aleksandra.stronska@pum.edu.pl (A.S.-P.); 3Department of Hygiene and Epidemiology, Collegium Medicum, University of Zielona Góra, 28 Zyty St., 65-045 Zielona Góra, Poland; chmiele@vp.pl (K.C.); chmiele1@tlen.pl (J.C.); 4Individual Specialized Medical Practice, Swarożyca 3/6 St., 71-601 Szczecin, Poland; smiarowskamalgorzatka@gmail.com; 5Foundation Strong in the Spirit, 60 Sienkiewicza St., 90-058 Łódź, Poland; health@mocniwduchu.pl

**Keywords:** OPRM1, gene, polymorphism, AUD, alcohol, women

## Abstract

The main aims of the present study were to explore the relationship of the OPRM1 gene rs1074287 polymorphism in alcohol-dependent women with their personality traits and to try to find out whether any specific features may influence alcohol cravings and be a prognostic for alcohol dependency and treatment in AUD women. Our study found a notable correlation between openness and the interaction of the ORIM1 gene and AUD. The alcohol use disorder subjects with genotype AG showed a higher level of openness compared to the control group with genotypes AG (*p* = 0.0001) and AA (*p* = 0.0125). The alcohol use disorder subjects with the AA genotype displayed higher levels of openness than the control group with genotype AG (*p* = 0.0271). However, the alcohol use disorder subjects with the AA genotype displayed lower levels of openness than the control group with genotype GG (*p* = 0.0212). Our study indicates that openness as a personality trait is correlated with the OPRM1 gene rs1074287 polymorphism in alcohol-dependent women. These are the first data and results exploring such a relationship between opioid and alcohol pathways and the mental construction of AUD women. Personality traits such as openness to experience and neuroticism might play major roles in the addiction mechanism, especially in genetically predisposed females, independent of the reward system involved in the emotional disturbances that coexist with anxiety and depression.

## 1. Introduction

Alcohol use disorder (AUD) is a multidimensional problem. It is caused by a combination of genetic and environmental components. Studies involving twins and adopted individuals have estimated that the genetic factor of AUD accounts for approximately 50% to 60% of its risk [1]. The endogenous opioid system is considered to play an important role in the development of excessive alcohol consumption, determining the risk of alcohol dependence due to its central role in the mesolimbic reward system, which is activated by both alcohol and opioids [2].

Alcohol possesses toxic and psychoactive properties that can lead to dependence. Its effects on the brain are complex, targeting multiple transmitter systems and exerting various effects on different aspects of the opioid signalling pathway. The suggested alterations in the opiate system concern both the subtypes of the receptors and the peptides in distinct brain areas, including the ventral striatum. Repeated ethanol administration has been observed to influence the production of prodynorphin, proopiomelanocortin, and pro-enkephalin [3] in specific areas of the rat brain, resulting in changes in the availability of natural ligands. Moreover, alcohol can induce structural alterations in peptide ligands that bind to opioid receptors or directly trigger the release of endorphins. Elevated levels of endogenous opioid ligands are proposed to modulate the pleasurable and reinforcing effects of alcohol, leading to alcohol dependence [4].

Opioid agonists and antagonists are known to impact not only alcohol consumption but also opioid medication, which have been shown to reduce alcohol intake. Therefore, they reduce the pleasurable experience and craving related to alcohol and can serve as a potential aversive method in the treatment of alcohol dependence [2].

Variation in the extracellular loops of opioid receptors regulates the interaction between the ligand and the receptor, allowing different degrees of specificity between various endogenous peptides and opioid receptor types [2]. Opioid receptors are G-protein-coupled receptors (GPCRs) belonging to the Rhodopsin family. These receptors activate downstream signalling by interacting with heterotrimeric G proteins. The most common types of opioid receptors are the μ-opioid receptor (MOR), δ-opioid receptor (DOR), and κ-opioid receptor (KOR) [5].

The β-endorphin and endomorphins are both cleavage products of the proopiomelanocortin precursor and can activate MOR; DOR is activated by enkephalin and dystrophin, while dynorphins activate the KOR receptor. These three receptor types are coded by the OPRM1, OPRD1, and OPRK1 genes, respectively [2].

Among the genes related to opioid receptors, OPRM1, which encodes the μ-opioid receptor, has been extensively investigated in the context of drug and alcohol dependence [6,7,8].

Through the whole-genome sequencing of various ethnic groups, researchers have discovered 3324 polymorphic sites in the OPRM1 gene, which is located in the sixth chromosome long arm and occupies an area of 200 kb. The majority of these polymorphisms occur at a very low frequency and are of little significance at the population level [2].

Numerous variations in the OPRM1 gene are associated with alcohol consumption and dependency, with Asn40Asp—rs1799971 being the most widely studied. A lower expression of the receptor has been observed in carriers of the G allele [9]. This variant has been repeatedly linked to different forms of dependency and the effectiveness of treatments for pain management.

A comprehensive genome-wide association study involving 10,000 subjects pinpointed a link between a specific SNP, known as rs1074287, which is situated 11.6 kilobases upstream of exon 1 within the OPRM1 gene, and the occurrence of heroin dependency [4,10]. This genetic polymorphism may also influence changes in libido levels and insomnia—the side effects that may occur in methadone maintenance treatment (MMT) patients [11].

Current research suggests that the degree of activation of the endogenous opioid system in response to alcohol dependence may, in part, be genetically determined. However, personality traits represent some of the most intricate and multifaceted quantitative characteristics of the AUD clinical pattern. Significant genetic covariance and associations were discovered between all personality traits and measures and distinctive symptoms of AUD, whereas enhancement motives were not significantly heritable [12]. For a long time, attention has been paid to the differences in and specificity of women’s alcohol use, resulting in a detailed model of their drinking, depth of alcohol abuse, social functioning, and health consequences, especially because of Foetal Alcohol Syndrome [13,14]. Women are more vulnerable than men in many aspects of the disease, despite the higher rates of substance use and use disorders (SUDs) among the latter. A striking example of this is the ‘telescoping effect’, which describes a faster progression of the disease in women when compared to men regarding the transition from the initiation of substance use to meeting the criteria for an SUD and/or seeking treatment. In relation to alcohol, this phenomenon was first observed more than 30 years ago [15]. Several subsequent studies replicated this observation [16], along with other classes of psychoactive substances, including stimulants (e.g., cocaine, nicotine/tobacco, and methamphetamine) [17]. The validity of the telescoping effect is also strongly supported by preclinical evidence showing that, as in humans, in animal studies, females are more likely than males to become dependent on psychoactive substances [18,19,20].

According to this analytical framework, many studies have consistently reported that women move faster from regular drinking to problem drinking and AUDs [21,22,23]. In addition, individuals with problematic use or AUD are more likely to seek treatment at an earlier stage if they are female than if they are male [24,25,26]. Given that women have a more severe clinical profile (e.g., more psychological, medical, social, and behavioural problems) at treatment entry than men, this more rapid progression to treatment seeking may be an effect of an earlier onset of severe SUD (five or more DSM-V symptoms) [27]. Studies indicating that women may experience an increased progression and/or susceptibility to alcohol-related health effects compared with men also support this conclusion. Health outcomes may differ, including a more rapid development of alcohol-related cirrhosis [28] and brain atrophy [29,30] in women than in men and greater alcohol-related effects on the heart and skeletal muscle in women than in men [31,32].

The main aims of the present study are to investigate the association of the OPRM1 gene rs1074287 polymorphism in alcohol-dependent women with their personality traits and to try to determine whether any specific characteristics may influence alcohol craving and be a prognostic in the case of alcohol dependence and the treatment of women with AUD. So far, evidence has been proven for this SNP and some other psychoactive substance use disorders (SUDs), such as the use of heroin. Much of the interest in AUD has focused on dopaminergic pathways, which are generally associated with pleasure, novelty seeking, and stress avoidance. At the same time, opioid regulation may support an antidepressant and anti-stress form of alcohol self-medication. The authors aim to present a model of selection and effective pharmacotherapy in the suffering of alcohol-dependent individuals. A step-by-step analysis of all types of receptors (MOR, DOR, and KOR) encoded by the OPRM1, OPRD1, and OPRK1 genes, which are selected from the set of genes documented in SUDs, especially in AUD women, is a very complicated task, but a necessary and unique one. One such task is the study of the rs1074287 polymorphism of the OPRM1 gene.

## 2. Results

The frequency distributions were consistent with the Hardy–Weinberg equilibrium (HWE) in both analysed groups (Table 1).

According to our study, there were no significant differences in the frequency of OPRIM rs1074287 genotypes between the alcohol use disorder subjects and the control group. The frequency of the A/A genotype was 0.47 in the alcohol use disorder subjects and 0.55 in the control group, while the frequency of the G/G genotype was 0.05 in the alcohol use disorder subjects and 0.09 in the control group. Additionally, the frequency of the A/G genotype was 0.49 in the alcohol use disorder subjects and 0.36 in the control group. The statistical analysis showed no significant difference between the two groups (χ^2^ = 4.084, *p* = 0.1298). Similarly, there was no significant difference in the frequency of individual alleles of the OPRIM rs1074287 polymorphism between the alcohol use disorder subjects and the control group. The frequency of the A allele was 0.71 in the alcohol-addicted group and 0.73 in the control group, while the frequency of the G allele was 0.29 in the alcohol use disorder subjects and 0.27 in the control group. The statistical analysis showed no significant difference between the two groups (χ^2^ = 0.310, *p* = 0.5823) (Table 2).

The means and standard deviations for the NEO-FFI traits of the alcohol use disorder subjects and control subjects are presented in Table 3.

The alcohol use disorder subjects, compared to the control group, obtained higher scores in the assessment of the NEO-FFI Neuroticism (7 vs. 5; Z = 8.287; *p* < 0.0001) and Openness (5 vs. 4.5; Z = 2.472; *p* = 0134) scales, and lower scores in the assessment of the NEO-FFI Extraversion (5 vs. 7; Z = −5.051; *p* < 0.0001), Agreeability (4 vs. 5; Z = −5.380; *p* < 0.0001), and Conscientiousness (5 vs. 7; Z = −5.844; *p* < 0.0001) scales (Table 3).

The results of a two-way ANOVA (OPRIM rs1074287 genotype and alcohol use disorder subjects or control group) of the NEO Five-Factor Personality Inventory (NEO–FFI) sten scales are summarised in Table 4.

There was a statistically significant interaction effect of the OPRIM rs1074287 genotype and alcohol use disorder subjects or control group on the Openness scale (F_2,207_ = 6.47 *p* = 0.0019; η^2^ = 0.059; Figure 1). The statistical power observed for this interaction of factors was 90%, and approximately 6% (η^2^) was explained by the OPRIM rs1074287 polymorphism and alcohol use disorder subjects or lack thereof on openness trait score variance.

A post hoc test revealed that the alcohol use disorder subjects with genotype AG showed a higher level of openness compared to the control group with genotypes AG (*p* = 0.0001) and AA (*p* = 0.0125). The alcohol use disorder subjects with the AA genotype displayed higher levels of openness than the control group with genotype AG (*p* = 0.0271). However, the alcohol use disorder subjects with the AA genotype displayed lower levels of openness than the control group with genotype GG (*p* = 0.0212). Finally, for the alcohol use disorder subjects with the GG genotype, there was no significant difference in openness levels to the control group with genotypes AG, AA, and GG. Furthermore, the control group with genotypes AG (*p* = 0.0003) and AA (*p* = 0.0068) displayed lower levels of openness than the control group with genotype GG. Table 5 shows the results of the post hoc test.

## 3. Discussion

The μ-opioid receptor (MOR) is a specific type of G-protein-coupled receptor that has a significant impact on reward and pleasure processes and has been associated with addictive behaviour [33]. This receptor has been found in the mesolimbic dopaminergic system of animal models, including in structures such as the basal ganglia and ventral tegmental area [34,35]. There is evidence suggesting that alcohol causes the release of endogenous opioids in the brain and indirectly activates opioid signalling pathways and that the rewarding effects and positive reinforcement experienced after alcohol consumption are regulated by opioid receptors [2,9,12].

The human opioid receptor gene OPRM1 has numerous polymorphisms within the exonic (coding) and intronic (non-coding) regions.

The OPRM1 gene has a particular location that is widely studied, which is a common missense single-nucleotide polymorphism (SNP) known as A118G rs1799971 [9]. This point mutation alters the protein’s amino acid sequence, and it has been found to have functional relevance in vitro, in vivo, and in human brain specimens. OPRM1 variants affect mu-opioid receptor binding and signalling, and neuropeptide gene expression levels related to addiction [36,37,38]. Most research suggests that individuals with the A118G SNP have reduced mu-opioid receptors [39].

The C17T variant in exon 1 of OPRM1 is a potentially functional variant that changes an alanine to a valine [40]. Studies have shown that rs1799972 is part of a haplotype that is associated with cocaine/heroin dependence in African Americans [41]. It has also been associated with quantitative drug abuse scores, specifically KMSK scales for cocaine, alcohol, and tobacco use in African American women [42]. However, some studies have shown negative findings [43].

Through a haplotype analysis, Shabalina et al. (2009) discovered a new genetic variant (rs563649) in the 5-prime untranslated region of the MOR1K isoforms that significantly contributed to pain sensitivity responses in 196 pain-free European American females. This C-T variant is located within an internal ribosome entry site (IRES) upstream of exon 13 and affects both the mRNA levels and translation efficiency of MOR1K isoforms. The study also found that rs563649 and rs1799971 were highly linked and that the minor T allele of rs563649 tagged a 6-SNP (AGTCTG) haplotype associated with high pain sensitivity [44].

Luo et al. (2003) conducted a study where they compared 318 European American individuals suffering from substance dependence (opioid and alcohol dependence) with 179 controls. The study found a significant difference in the haplotype frequency of alleles at the OPRM1 locus, with a *p*-value of 0.0036. The allele -2044C-A in the 5-prime putative regulatory region and haplotypes, including -2044C-A, were identified as the susceptibility allele and haplotypes, respectively, associated with substance dependence [45].

Three intronic SNPs (rs495491, rs6091485, and rs648893) were associated with alcohol dependence in a study of European Americans [13].

The OPRM1 gene has a polymorphism identified by the rs1074287 marker. This variant is located in the intron region and is characterised by an A/G change at the Chr 6:154027674 position. A genome-wide association study of 10,000 subjects identified a link between a specific SNP, rs1074287, and heroin addiction. This SNP is located 11.6 kilobases upstream of exon 1 within the OPRM1 gene [4,10]. The main aims of the present study were to explore the relationship of the OPRM1 gene rs1074287 polymorphism in alcohol-dependent women with their personality traits and to try to find out whether there are any specific features that may influence alcohol cravings and be a prognostic for alcohol dependency and treatment in AUD women.

According to our study, there was no significant difference in the frequency of OPRIM rs1074287 genotypes between the alcohol use disorder subjects and the control group. The frequency of the A/A genotype was 0.47 in the alcohol use disorder subjects and 0.55 in the controls; the frequency of the G/G genotype was 0.05 and 0.09 in the alcohol use disorder subjects and the controls, respectively; and the frequency of the A/G genotype was 0.49 in the alcohol use disorder subjects and 0.36 in the controls. The statistical analysis showed no significant difference between these two analysed groups (χ^2^ = 4.084, *p* = 0.1298). Similarly, there was no significant difference in the frequency of individual alleles of the OPRIM rs1074287 gene between the alcohol use disorder subjects and the control group. A comparison of the frequency of the A allele found that it was 0.71 in the alcohol use disorder subjects and 0.73 in the control group, while the frequency of the G allele was 0.29 in the alcohol use disorder subjects and 0.27 in the control group. The statistical analysis showed no significant difference between the examined alcohol-dependent and control women (χ^2^ = 0.310, *p* = 0.5823).

Research conducted on post-mortem brain tissue samples of individuals who suffered from depression and died by suicide revealed a correlation between MOR protein levels and the frequency of the rs1074287 allele. The study found that individuals who carried the minor G allele had higher levels of the MOR protein, which resulted in a greater potency of the DAMGO, a mu-opioid receptor selective agonist (*p* = 0.017) [14].

Certain variants in the OPRM1 gene may affect how individuals respond to rewards and stimuli, which, in turn, can impact their chances of developing alcohol dependency. If an individual possesses a specific genotype variant that makes them experience a heightened reward effect when consuming alcohol, it could increase their susceptibility to dependency.

However, it is important to point out that the OPRM1 gene is not the only determining factor in alcohol dependence. The phenomenon of addiction is complex, and it can be affected by a range of factors, such as genetics, environment, psychology, and social influences.

Although certain genetic associations may increase the risk of alcohol dependence, it is not the only decisive factor. Individuals at risk of alcohol dependency can benefit from different forms of support and therapy that consider the comprehensive context of this issue.

Therefore, a meta-analysis was conducted to explore the relationship between alcohol consumption and personality traits measured by the widely used Five-Factor Model, which includes neuroticism, extraversion, conscientiousness, openness to experience, and agreeableness. The cohort’s age, sex, and race were adjusted for in the analysis. The results showed that subjects with higher levels of extraversion and lower levels of conscientiousness were characterised by an increased risk of developing moderate-to-heavy alcohol consumption over time. However, individuals with lower levels of extraversion, higher levels of agreeableness, and lower levels of openness were prone to an increased likelihood of reducing alcohol consumption or choosing abstinence. These findings suggest that people who might be less open to experience and more agreeable are predisposed to prefer abstinence, while those who are extroverts and of low conscientiousness tend to be more prone to increased alcohol consumption [33].

Research has shown that people prone to negative emotions, as measured by the Neuroticism, Extraversion, and Openness Personality Inventory (NEO-PI) Neuroticism scale, are more likely to develop major depression. In addition, a serotonin transporter promoter polymorphism called 5-HTTLPR has been associated with higher neuroticism scores in several psychiatric disorders. It has also been found to increase the likelihood of alcohol use. Similarly, both alcohol and depression-related traits have been associated with the GABA(A) receptor alpha6 subunit [34].

Our study found that the alcohol use disorder subjects with genotype AG showed a higher level of openness compared to the control group with genotypes AG (*p* = 0.0001) and AA (*p* = 0.0125). The alcohol use disorder subjects with the AA genotype displayed higher levels of openness than the control group with genotype AG (*p* = 0.0271). However, the alcohol use disorder subjects with the AA genotype displayed lower levels of openness than the control group with genotype GG (*p* = 0.0212). Finally, for the alcohol use disorder subjects with the GG genotype, there was no significant difference in openness levels to the control group with genotypes AG, AA, and GG.

A significant effect of the OPRIM rs1074287 genotype and group interaction was observed. This effect was observed on the Openness scale for both the AUD and control groups (F_2,207_ = 6.47, *p* = 0.0019, η^2^ = 0.059).

Our results of a test comparing individuals who are alcohol-dependent to a control group also showed that the former scored higher in the assessment of the Neuroticism scale of NEO-FFI (7.26 vs. 4.55; Z = 8.287; *p* < 0.0001) and the Openness scale of NEO-FFI (5.11 vs. 4.50; Z = 2.472; *p* = 0.134). However, the AUD group scored lower on the Extraversion scale of NEO-FFI (5.11 vs. 6.72; Z = −5.051; *p* < 0.0001), the Agreeability scale of NEO-FFI (3.84 vs. 5.49; Z = −5.380; *p* < 0.0001), and the Conscientiousness scale of NEO-FFI (4.97 vs. 6.88; Z = −5.844; *p* < 0.0001).

It has been shown that openness to experience is associated with certain genotypes and alleles of ADH1C and ALDH2 [35], but there have been no data about the correlation between the OPRM1 gene and personality traits up to now. Our data presented above are the first to prove that openness to experience may be related to OPRM1 gene polymorphisms.

These findings also seem to be congruent with the personality configuration, according to the biopsychosocial model of personality developed by American psychiatrist and psychologist Robert C. Cloninger, which suggests that personality may be a risk factor that predisposes an individual to develop alcoholism in response to stress and challenges. Cloninger divided alcoholism into two types [46,47]. Type I, which is more common in women, is caused by environmental factors and typically affects people with neurotic, schizoid, passive, perfectionistic, and depressive personality traits. Addiction develops quickly, and long periods of drinking with long periods of abstinence are typical. Individuals with this type of alcoholism are more likely to suffer from guilt, depression, and other mental and somatic complications.

Type II, however, is caused by biological factors and is usually passed down from father to son. It is characterised by impulsive, extroverted, and antisocial personality disorders belonging to Cluster B Personality Disorders (according to DSM-IV) [48]. People with type II alcoholism tend to initiate alcohol use in adolescence, and the periods of excessive drinking are long and continuous. This type of alcoholism is more socially accepted, and, therefore, the criticism towards drinking and its consequences is less [49,50].

As a result, it can be said that drinking in men is spontaneous, while in women, it is often secondary and conditioned by various factors. Women may use alcohol to cope with stress or avoid negative experiences, while men are more pleasure-seeking and prone to seeking rewarding stimuli that increase the risk of behavioural addiction, among which alcohol is one of the most available solutions [51].

The differences in alcohol drinking patterns between women and men are not only due to cultural, environmental, metabolic, and hormonal differences but also to sensitivity to alcohol’s effects, which depend on personality traits and brain biochemistry.

It has been observed that the combination of certain personality traits, such as impulsivity, affective instability, and negative affectivity, which are specific to Cluster B Personality Disorders [52], may increase the risk of alcohol dependence. These traits are also known to contribute to symptoms of anxiety and depression disorders by regulating emotional states, especially negative ones. Although the analysis of the female model of drinking, adjusting for coping motives, showed that genetic correlations between AUD symptoms and traits of neuroticism and agreeableness might be expected, this has not yet been exactly proved.

All these disorders, including anxiety, depression, and behavioural disturbances, are extremely maladaptive variants of normal personality traits. Therefore, understanding the associations between personality traits, genetic background, and AUD can help identify specific features that contribute to these mental disorders and/or disturbances.

## 4. Materials and Methods

### 4.1. Participants

The study group included 213 female subjects. Of these, 101 were diagnosed with alcohol use disorder (mean age = 45.74, SD = 11.11), and 112 were not alcohol-dependent (control group; mean age = 45.32, SD = 10.19). The study was previously approved by the Bioethics Committee of the District Medical Council in Zielona Góra (KB-07/72/2017). Written informed consent was obtained from all participants, and the study was conducted at the Independent Laboratory of Health Promotion, Pomeranian Medical University, in Szczecin. Individuals with AUD and the control group were assessed by psychiatrists using the Mini-International Neuropsychiatric Interview (MINI), and both groups completed the NEO-FFI questionnaire independently.

### 4.2. Psychometric Measures

The Mini-International Neuropsychiatric Interview is a structured diagnostic interview used to evaluate psychiatric diagnoses according to DSM-IV and ICD-10 criteria. The Personality Inventory (NEO-FFI Five-Factor Inventory, NEO-FFI) contains six components for each of the five traits—neuroticism (anxiety, hostility, depression, self-awareness, impulsivity, susceptibility to stress), extroversion (warmth, sociability, assertiveness, activity, emotion seeking, positive emotions), openness to experience (fantasy, aesthetics, feelings, actions, ideas, values), agreeableness (trust, straightforwardness, altruism, compliance, modesty, tenderness), and conscientiousness (competence, order, duty, striving for achievements, self-discipline, consideration) [53].

The results of the NEO-FFI are reported as sten scores. The conversion of raw scores to sten scores was carried out by following the Polish standards for adults, where it was assumed that 1–2 corresponded to very low results, 3–4 corresponded to low results, 5–6 corresponded to average results, 7–8 corresponded to high results, and 9–10 corresponded to very high results.

### 4.3. Genotyping

Genomic DNA was isolated from venous blood by using standard procedures. Genotyping was conducted with the real-time PCR method. The fluorescence signal was plotted as a function of temperature to provide melting curves for each sample. The OPRM1 gene rs1074287 peaks were read at 61.68 °C for the A allele and 68.33 °C for the G allele.

### 4.4. Statistical Analysis

Concordance between the genotype frequency distribution and Hardy–Weinberg equilibrium (HWE) was tested using HWE software (https://wpcalc.com/en/equilibrium-hardy-weinberg/ (accessed on 1 April 2023). The NEO Five-Factor Inventory (comprising Neuroticism, Extraversion, Openness, Agreeability, and Conscientiousness scales) was not normally distributed; therefore, comparisons of inventory items between genotypes used the Mann–Whitney U-test. When assessing the relationship between OPRIM rs1074287 variants, AUD and control subjects, and the NEO Five-Factor Inventory, a two-way ANOVA model was used, as the assumption of homogeneity of variance was fulfilled (Levene test *p* > 0.05). The ANOVA model included the main effects of the OPRIM variant and AUD/control status, as well as the interaction between the two for each inventory outcome. All computations were performed using STATISTICA 13 (Tibco Software Inc., Palo Alto, CA, USA) for Windows (Microsoft Corporation, Redmond, WA, USA).

## 5. Conclusions

Our study suggests that openness as a personality trait is correlated with the OPRM1 gene rs1074287 polymorphism in alcohol-dependent women. These are the first data and results exploring such a relationship between opioid and alcohol pathways and the mental construction of alcohol-dependent women. Personality traits such as openness to experience and neuroticism might play major roles in substance dependency mechanisms, especially in genetically predisposed females, independent of the reward system involved in the emotional disturbances that coexist with anxiety and depression.

## Figures and Tables

**Figure 1 ijms-25-03067-f001:**
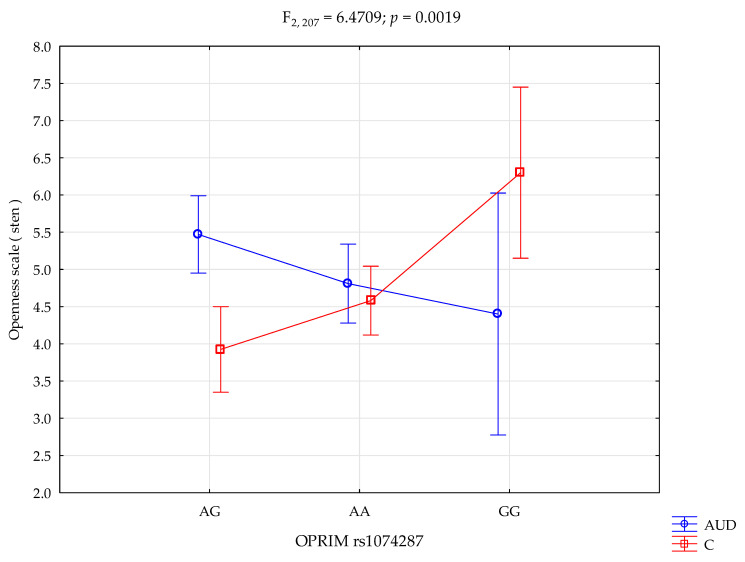
Interaction between the alcohol use disorder (AUD) subjects/control (C) and OPRIM rs1074287 and Openness scale. Means and standard errors are presented.

**Table 1 ijms-25-03067-t001:** Hardy–Weinberg equilibrium for alcohol use disorder (AUD) subjects and control subjects.

Hardy–Weinberg Equilibrium, Including Analysis of Ascertainment Bias		Observed (Expected)	Allele Freq	χ^2^ (*p*-Value)
OPRIM rs1074287				
AUD *n* = 101	A/A	47 (50.6)	p (A)= 0.71 q (G)= 0.29	3.029 (0.0818)
	A/G	49 (41.8)		
	G/G	5 (8.6)		
Control *n* = 112	A/A	62 (60.0)	p (A)= 0.73 q (G)= 0.27	0.896 (0.3439)
	A/G	40 (43.9)		
	G/G	10 (8.0)		

*p*—statistical significance χ^2^ test.

**Table 2 ijms-25-03067-t002:** Frequency of genotypes and alleles of the OPRIM gene rs1074287 polymorphisms in the alcohol use disorder (AUD) subjects and control subjects.

OPRIM rs1074287					
		Genotypes		Alleles	
	A/A *n*(%)	A/G *n*(%)	G/G *n*(%)	A *n*(%)	G *n*(%)
AUD *n* = 101	47 (46.53%)	49 (48.51%)	5 (4.95%)	143 (70.79%)	59 (29.21%)
Control N = 112	62 (55.36%)	40 (35.71%)	10 (8.93%)	164 (73.21%)	60 (26.79%)
χ^2^ (*p*-value)		4.084 0.1298		0.310 (0.5823)	

*n*—number of subjects.

**Table 3 ijms-25-03067-t003:** NEO Five-Factor Inventory sten scores for alcohol use disorder (AUD) subjects and control group.

NEO Five-Factor Inventory	AUD (*n* = 101) Q1; Q2; Q3	Control (*n* = 112) Q1; Q2; Q3	(*p*-Value)
Neuroticism scale	6; 7; 8	3; 5; 6	<0.0001 *
Extraversion scale	4; 5; 7	5.5; 7; 8	<0.0001 *
Openness scale	4; 5; 7	4; 4.5; 5.5	0.0134 *
Agreeability scale	3; 4; 5	4; 5; 7	<0.0001 *
Conscientiousness scale	3; 5; 7	6; 7; 8	<0.0001 *

*p*—statistical significance with Mann–Whitney U-test; Q1, Q2, Q3—interquartile range; *n*—number of subjects; *—statistically significant differences.

**Table 4 ijms-25-03067-t004:** The results of two-way ANOVA for alcohol use disorder (AUD) subjects and controls, OPRIM rs1074287, and NEO Five-Factor Inventory.

NEO Five- Factor Inventory	Group	OPRIM rs1074287			Two-Way ANOVA			
		A/A *n* = 109 M ± SD	A/G *n* = 89 M ± SD	G/G *n* = 15 M ± SD	Factors	*p*-Value	ɳ^2^	Statistical Power (Alfa = 0.05)
Neuroticism scale	Alcohol use disorder (AUD): *n* = 101	7.30 ± 1.94	7.20 ± 1.82	7.40 ± 1.67	AUD/control rs1079597 AUD/control × rs1079597 #	*p* < 0.0001 * *p* = 0.7832 *p* = 0.6183	0.169 0.002 0.005	0.999 0.088 0.128
	Control: *n* = 112	4.35 ± 2.09	4.82 ± 2.18	4.70 ± 1.77				
Extraversion scale	Alcohol use disorder (AUD): *n* = 101	4.72 ± 2.36	5.41 ± 2.00	5.80 ± 3.27	AUD/control rs1079597 AUD/control × rs1079597 #	*p* = 0.0009 * *p* = 0.2492 *p* = 0.6225	0.052 0.013 0.005	0.919 0.298 0.127
	Control: *n* = 112	6.63 ± 1.79	6.85 ± 2.03	6.80 ± 2.35				
Openness scale	Alcohol use disorder (AUD): *n* = 101	4.81 ± 2.26	5.47 ± 2.02	4.40 ± 1.82	AUD/control rs1079597 AUD/control × rs1079597 #	*p* = 0.9110 *p* = 0.4576 *p* = 0.0019 *	0.0001 0.008 0.059	0.051 0.182 0.902
	Control: *n* = 112	4.58 ± 1.50	3.92 ± 1.44	6.30 ± 2.16				
Agreeability scale	Alcohol use disorder (AUD): *n*= 101	3.87 ± 1.69	3.88 ± 2.08	3.20 ± 1.48	AUD/control rs1079597 AUD/control × rs1079597 #	*p* < 0.0001 * *p* = 0.3789 *p* = 0.2082	0.104 0.009 0.015	0.998 0.218 0.333
	Control: n = 112	5.11 ± 2.28	5.92 ± 2.00	6.10 ± 1.97				
Conscientiousness scale	Alcohol use disorder (AUD): *n* = 101	5.21 ± 2.39	4.82 ± 2.31	4.20 ± 2.17	AUD/control rs1079597 AUD/control × rs1079597 #	*p* < 0.0001 * *p* = 0.6495 *p* = 0.6466	0.097 0.004 0.004	0.997 0.120 0.120
	Control: *n* = 112	6.87 ± 1.96	6.92 ± 1.84	6.80 ± 2.94				

*—significant result; AUD—alcohol use disorder subjects; M ± SD—mean ± standard deviation; ɳ^2^—a measure of the strength of the effect (ranges from 0 to 1; this indicator shows what percentage of variability in the NEO Five-Factor Inventory features is explained by the alcohol addiction and control variables and by the OPRIM rs1074287 gene polymorphism); #—interaction between OPRIM rs1074287 genotype and alcohol use disorder subjects or control group (AUD/control × rs1079597).

**Table 5 ijms-25-03067-t005:** Post hoc (Least Significant Difference) analysis of interactions between the alcohol use disorder (AUD) subjects/control and OPRIM rs1074287 and Openness scale.

	OPRIM rs1074287 and Openness Scale					
	AUD A/A M = 4.81	AUD A/G M = 5.47	AUD G/G M = 4.40	Control A/A M = 4.58	Control A/G M = 3.92	Control G/G M = 6.30
AUD A/A		0.0808	0.6384	0.5238	0.0271 *	0.0212 *
AUD A/G			0.2184	0.0125 *	0.0001 *	0.1959
AUD G/G				0.8334	0.5879	0.0615
Control A/A					0.0812	0.0068 *
Control A/G						0.0003 *
Control G/G						

*—significant statistical differences, M—mean.

## Data Availability

The genotyping and psychometric test results are available upon request.

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
