# Peer review of "OPRM1 Gene Polymorphism in Women with Alcohol Use Disorder"

_ijms, 2024, doi:10.3390/ijms25053067_

Round 1

Reviewer 1 Report

Comments and Suggestions for Authors

As I read the manuscript, I had a very difficult time following the results as presented, therefore, it is difficult to review the soundness of the conclusions and whether they are supported by the data.  A few specific comments include:

The statistical method section describes the analysis as “relations between OPRIM rs1074287 variants Addicted to alcohol and control subjects and the NEO Five-Factor Inventory were analysed using a multivariate analysis of factor effects ANOVA [NEO-FFI scale as dependent variable, genetic feature, group and interaction genetic feature × group as independent variables.”  The results sections describe the analysis as a “2x3 factorial ANOVA”.  Why the use of different terminology?  It would be much easier for the general reader if the same terminology was used throughout.

The presentation of the ANOVA model is very confusing.  A 2X3 factorial ANOVA tests three different hypotheses. A main effect of each independent variable (alcohol/control and genotype here) and an interaction between those two variables.  For each inventory scale outcome, table 4 reports 4 f-tests but it is not clear what any of them refer to?  Where are the main effects and the interactions reported?  And if there is a significant interaction (which it looks like there is from figure 1), were differences in the dependent variable with respect to one of the factors compared post-hoc and what were those results?  Table 5 is labeled post-hoc tests but I do not understand what it is showing.  For example: the addicted to alcohol A/G group shows a value of 0.0001* under openness level 4.  What is that statistically significantly different from?

In addition, I’m not sure what the inclusion of a power estimate for each F-test in table 4 is for and the effect size column needs to be explained as most readers will not know what the “n2” represents?  Neither effect size nor power are mentioned anywhere in the manuscript.

Two minor comments:

1     The study is described as “temporary” (Page 3, Line 119).  What is the definition of temporary here?

 In table 1, genotype frequencies with HWE testing are shown.  However, why do the authors order the genotypes as heterozygote, homozygote wildtype, and homozygote mutant?  It is an unusual order to report genotype frequencies.

Author Response

Answer to the reviewer 1

We sincerely thank you for all Your very valuable comments. Changes were made in the manuscript, which can be seen in the changes tracking panel.

  • The statistical method section describes the analysis as “relations between OPRIM rs1074287 variants Addicted to alcohol and control subjects and the NEO Five-Factor Inventory were analysed using a multivariate analysis of factor effects ANOVA [NEO-FFI scale as dependent variable, genetic feature, group and interaction genetic feature × group as independent variables.” The results sections describe the analysis as a “2x3 factorial ANOVA”.  Why the use of different terminology?  It would be much easier for the general reader if the same terminology was used throughout.

The presentation of the ANOVA model is very confusing.  A 2X3 factorial ANOVA tests three different hypotheses. A main effect of each independent variable (alcohol/control and genotype here) and an interaction between those two variables.  For each inventory scale outcome, table 4 reports 4 f-tests but it is not clear what any of them refer to?  Where are the main effects and the interactions reported?  And if there is a significant interaction (which it looks like there is from figure 1), were differences in the dependent variable with respect to one of the factors compared post-hoc and what were those results?  Table 5 is labeled post-hoc tests but I do not understand what it is showing.  For example: the addicted to alcohol A/G group shows a value of 0.0001* under openness level 4.  What is that statistically significantly different from?

In addition, I’m not sure what the inclusion of a power estimate for each F-test in table 4 is for and the effect size column needs to be explained as most readers will not know what the “n2” represents?  Neither effect size nor power are mentioned anywhere in the manuscript.

Thank you for the comment. We improved our data reporting - two-way ANOVA term was used. The description of the two-way ANOVA results has been improved. Interactions are described in Table 4 using the # sign.

The meaning of ɳ2 is described in Table 4, and the fragment that concerns it is marked in the text. The description of the text regarding the Post hoc test has been corrected. The entry in Table 5 has been corrected.

  • The study is described as “temporary” (Page 3, Line 119). What is the definition of temporary here?

Thank you for your perceptiveness and attention. It was a language-related mistake of non-native speakers. “Temporary” was corrected to “present”.

  • In table 1, genotype frequencies with HWE testing are shown.  However, why do the authors order the genotypes as heterozygote, homozygote wildtype, and homozygote mutant?  It is an unusual order to report genotype frequencies.

Thank you for your perceptiveness and attention. The order of homozygote wildtype, heterozygote and homozygote mutant has been corrected throughout the manuscript.

Reviewer 2 Report

Comments and Suggestions for Authors

Authors should convince readers for their selection rationale. Starting from abstract authors should talk about why they focused that specific SNP what is outcome, how their results will help for further studies or current knowledge. 

Authors expand their statistical analysis and provide more data (Demographics, more correlation relevant with their study)

Authors should provide more examples about mutation (What is already known?)

Author Response

Answer to the reviewer 2

We sincerely thank you for all Your valuable comments. Below is the location of all changes, which can additionally be seen in the changes tracking panel.

  • Authors should convince readers for their selection rationale. Starting from abstract authors should talk about why they focused that specific SNP what is outcome, how their results will help for further studies or current knowledge.

Thank You for the suggestion. The information was added in lines 21-25 and 171-185.

  • Authors expand their statistical analysis and provide more data (Demographics, more correlation relevant with their study)

Thank you for the comment. Alcohol-dependent women are a very difficult group to cooperate with for researchers. It was a big problem to obtain any reliable and detailed information from them, which is why the article does not include demographic data, which we sincerely regret.

  • Authors should provide more examples about mutation (What is already known?)

Thank You for making the improvement. The new examples were added in lines 294-319.

Round 2

Reviewer 1 Report

Comments and Suggestions for Authors

The manuscript is much improved and easier to follow.  There are a few areas where I would make minor changes and I have put suggestions together below.

Minor comments:

Table 2: I would recommend removing the Z-values as they don’t really add much information to the average reader and make the tables more complicated than necessary.  In addition, because the NEO outcomes were compared using a nonparametric Mann-Whitney test, I would report the medians and IQR in place of the means and standard deviations.

Table 3: I would recommend removing the F-values for the same reason listed above.  In addition, I would remove the intercept term from the tables.  We rarely report the intercept, and I think that having 4 p-values for each model was confusing to me the first time I reviewed the manuscript as I wasn’t expecting the intercept.  With only reporting the three factors, it is clearer that you are reporting the two main effects and the interaction.

I would recommend that the second through fourth sentence of the statistical analysis methods section be changed slightly to say:

The NEO Five-Factor Inventory (Neuroticism, Extraversion, Openness, Agreeability, and Conscientiousness) were not normally distributed, therefore comparisons of inventory items between genotypes used the U Mann-Whitney test.  When assessing the relationship between OPRIM rs1074287 variants, AUD and control subjects, and the NEO Five Factor Inventory, a two-way ANOVA model was used as the assumption of homogeneity of variance was fulfilled Levene test p > 0.05). The ANOVA model included main effects of OPRIM variant and AUD/control status, and the interaction between the two for each inventory outcome.

This, or similar language, makes it clearer what was done and lists the methods used in the same order they are reported in the manuscript.

Author Response

Answer to the reviewer

We sincerely thank you for all your comments. Below is the location of all changes, which can be seen in the changes tracking panel.

Comments and Suggestions for Authors

The manuscript is much improved and easier to follow.  There are a few areas where I would make minor changes and I have put suggestions together below.

Minor comments:

Table 2: I would recommend removing the Z-values as they don’t really add much information to the average reader and make the tables more complicated than necessary.  In addition, because the NEO outcomes were compared using a nonparametric Mann-Whitney test, I would report the medians and IQR in place of the means and standard deviations.

Thank you for the suggestion; the table was rearranged (Table 3, line 165).

Table 3: I would recommend removing the F-values for the same reason listed above.  In addition, I would remove the intercept term from the tables.  We rarely report the intercept, and I think that having 4 p-values for each model was confusing to me the first time I reviewed the manuscript as I wasn’t expecting the intercept.  With only reporting the three factors, it is clearer that you are reporting the two main effects and the interaction.

Thank you for the suggestion; the table was rearranged (Table 4, line 198).

I would recommend that the second through fourth sentence of the statistical analysis methods section be changed slightly to say:

The NEO Five-Factor Inventory (Neuroticism, Extraversion, Openness, Agreeability, and Conscientiousness) were not normally distributed, therefore comparisons of inventory items between genotypes used the U Mann-Whitney test.  When assessing the relationship between OPRIM rs1074287 variants, AUD and control subjects, and the NEO Five Factor Inventory, a two-way ANOVA model was used as the assumption of homogeneity of variance was fulfilled Levene test p > 0.05). The ANOVA model included main effects of OPRIM variant and AUD/control status, and the interaction between the two for each inventory outcome.

 This, or similar language, makes it clearer what was done and lists the methods used in the same order they are reported in the manuscript.

Thank you for the comment; the information was added (lines 409- 417).

 We sincerely thank you for your valuable comments on the article and your significant contribution to improving it.

Reviewer 2 Report

Comments and Suggestions for Authors

Authors remarkably improved manuscript.

They nearly re-write whole manuscript and in that format it is suitable for further steps.

No comments needed.

Author Response

Answer to the Reviewer

Comments and Suggestions for Authors

Authors remarkably improved manuscript.

They nearly re-write whole manuscript and in that format it is suitable for further steps.

No comments needed.

Dear Reviewer,

We sincerely thank you for your valuable comments on the article and your significant contribution to improving it.